# Strategies for Feeding Unweaned Dairy Beef Cattle to Improve Their Health

**DOI:** 10.3390/ani10101908

**Published:** 2020-10-18

**Authors:** Maria Devant, Sonia Marti

**Affiliations:** Ruminant Production, IRTA, Torre Marimon, 08140 Caldes de Montbui, Spain; sonia.marti@irta.cat

**Keywords:** unweaned calves, nutrition, gut health, bovine respiratory disease

## Abstract

**Simple Summary:**

In intensive fattening cattle, little attention has been focused to discuss if nutritional strategies could help reducing the incidence and severity of the most important health disease in feedlot cattle: the bovine respiratory disease (BRD). Bovine respiratory disease has a great impact on animal productivity with great morbidity and mortality rates during the first months after arrival. Metaphylactic antimicrobial programs are used to prevent and treat it, however, this strategy fails in aspects of the prudent use of antimicrobial treatments promoting the selection for resistance gene determinants and antimicrobial-resistant bacteria. The present review tries to answer the question if there are any nutritional strategies that could help to reduce the incidence and severity of BRD in dairy beef calves. Dairy beef calves are the calves born in dairy farms, around 65%, that are not going to replace the dairy cows and are fattened and slaughtered for meat production. These unweaned and unwanted calves are not a priority for the dairy sector and as a result, their postnatal care has not been a priority for them, and they are considered a by-product. Currently improving their health and welfare is a worldwide concern.

**Abstract:**

In order to answer the question of whether nutritional interventions may help to reduce the incidence of respiratory disease in dairy beef calves at arrival, the present review is divided in three sections. In the first section, the nutrition of calves previous to the arrival from the origin farm to the final rearing farm is reviewed. In the second section, the possible consequences of this previous nutrition on gut health and immune status upon arrival to the rearing farm are described. The main consequences of previous nutrition and management that these unweaned calves suffer at arrival are the negative energy balance, the increased intestinal permeability, the oxidative stress, the anemia, and the recovery feed consumption. Finally, in the third section, some considerations to advance in future nutritional strategies are suggested, which are focused on the prevention of the negative consequences of previous nutrition and the recovery of the gut and immune status. Moreover, additional suggestions are formulated that will be also helpful to reduce the incidence of bovine respiratory disease (BRD) that are not directly linked to nutrition like having a control golden standard in the studies or designing risk categories in order to classify calves as suitable or not to be transported.

## 1. Introduction

In intensive cattle fattening, the major diseases related to nutrition and feeding programs are rumen acidosis and bloat [1,2], although a definitive diagnosis under commercial conditions is seldom secured. Dietary strategies such as feeding management and the supplementation of feed additives have been adopted to reduce the risk of suffering them [3]. In recent decades, there has been a big effort to study the different factors that affect the onset and severity of rumen acidosis and very extensive reviews have been published [2,4]. However, little attention has been focused to discuss if nutritional strategies, among other management approaches, could help to improve the cattle health status. The most important health disease in feedlot cattle is the bovine respiratory disease (BRD) [1,5]. Viruses and stress-related behaviors interfere with the mucociliary balance of the respiratory tract and dysregulate the tracheal antimicrobial peptides of the innate defense, allowing for viral and bacterial pathogens to replicate within the respiratory tract [6]. BRD accounts for 70% to 80% of all morbidity and from 40% to 50% of all mortality in beef cattle [7] and estimated costs to the American feedlot industry is around USD 800 million to 900 million annually [8,9]. Most BRD outbreaks and number of treatments likely happen in the first 2 w of arrival and calf mortality in the first month after arrival [7,10]. Timsit et al. [11] observed that fever episodes occurred frequently during the first weeks after entrance and they can last up to 11 days. BRD is also known as “shipping fever” since it occurs soon after the cattle arrive at the feedlot and the stress from shipping is considered one of the major factors producing the disease [12]. Recently, using thoracic ultrasound as a tool for BRD, Tejero et al. [13] observed that 20% of the calves coming from auction markets and concentration centers had lung lesions at arrival, and the percentage of those lesions increased to 76% within 19 d. In contemporary high-intensity beef production, metaphylactic antimicrobial programs continue to be a management option at the entrance of calves to prevent and treat BRD and this strategy fails in aspects of the prudent use of antimicrobial treatments, like the need of having an accurate diagnosis before deciding for the treatment with antimicrobials [14,15]. Still many different antimicrobials are used injudiciously with no real rationale promoting the selection for resistance gene determinants and antimicrobial-resistant bacteria [16,17]. These antimicrobial-resistant pathogens can complicate the prevention and treatment of infectious diseases in beef farms. Furthermore, the transmission of antimicrobial resistance genes to bovine-associated human pathogens is a potential public health concern. To date, in beef production the use of metaphylactic and prophylactic antimicrobials masks deficient management practices and prevents the increase in morbidity and mortality. The question is: are there any nutritional strategies that could help to reduce the incidence and severity of BRD? How could nutrition, or the digestive tract health, be linked to BRD? Duff and Galyean [5] conducted an excellent review analyzing different nutritional strategies as possible strategies to reduce BRD incidence. Moreover, in the last 10 years, the gastrointestinal tract (GIT) and its health status have received increasing attention in the commercial and scientific community. The term “gut health” describes the optimal conditions of the GIT such as an effective digestion and absorption of nutrients, absence of disease, a normal and stable intestinal microbiota, and an effective immune status. Furthermore, Budden at al. [18] summarized a new paradigm in their paper “Emerging pathogenic links between microbiota and the gut–lung axis”. Recently, Enaud et al. [19] reviewed different crosstalk mechanisms in humans of how lung and gut microbiota can influence each other and how they interact with the immune system and their role in the respiratory system diseases. These authors describe two main pathways: first, the mesenteric lymphatic system through which intact bacteria, their fragments or metabolites (short-chain fatty acids, SCFA) may translocate across the intestinal barrier and reach the systemic circulation and modulate the lung immune response. Second, the gut segmented filamentous bacteria, that are commensal bacteria of the ileum, regulate CD4+ T-cell polarization on the Th17 pathway, which is implicated in the response to pulmonary fungal infections and lung autoimmune pathologies. Interestingly, these authors mention several studies where the use of broad-spectrum antibiotic treatment disrupts the gut microbiota and reduces the response against lung infections. In ruminants, there is a gap in this research area and this area could open a new opportunity to study the link between the gut–lung axis and all stressors that those animals suffer from during their productive life. In our professional life, we have heard very often from the farmers that “after a digestive disorder or a stress event we observe in the coming days a BRD outbreak”. Thus, the understanding of this possible crosstalk and the main factors altering it may help us in the future to design nutritional strategies to reduce the BRD severity and incidence improving animal welfare, antibiotic use, and therefore beef production sustainability. 

To answer these questions, let us first focus on the nutrition and factors (like stress) that may modify their nutritional requirements around the BRD onset. The questions we need to answer are: how does the life of those calves look like before they arrive? Calf marketing and transport is part of the beef industry and necessary as a part of the production cycle, involving practices like the sorting and mixing/commingling in auction markets, feed restriction or fasting periods. All these practices around transport have physiological consequences like dehydration, difficulties to thermoregulate, body weight (BW) losses, tissue damage, feed deprivation, smoke inhalation that cause stress, inflammation and immune suppression [20]. Thus, the production system by itself has an important predisposing factor to suffer respiratory disease. When looking into intensive beef production we have two different scenarios: the transportation of unweaned calves—around 1 month of age (dairy beef calves), and the transportation of weaned calves (feedlot cattle) around 6 months of age. These two different scenarios have several points in common, but they differ in other points, for example, a 6 month-old calf has a rumen fully developed and a mature immune system and has probably been castrated, dehorned, and vaccinated before animal transport. Wilson et al. [21] did an excellent review around management practices before and after arrival to health and animal well-being in newly weaned calves—around 6 months of age. The objectives of the present review will be to analyze the potential nutritional strategies for unweaned dairy beef cattle to improve their health linked to gut—lung axis. To understand the potential of nutrition in improving the health status of dairy beef calves it is important to understand and better describe the dairy beef calf’s production system.

## 2. The Dairy Beef Calves

Most of the unweaned calves that arrive to the rearing farms are male calves from dairy production which have for long been considered a by-product or even a waste product. In Europe, the dairy sector is one of the largest agricultural sectors in the EU, representing more than 13.2% of total agricultural output [22]. Total milk production in the EU is estimated around 172 million tonnes per year [22], 167 million tonnes per year correspond to cow’s milk. The EU’s main milk producers are Germany (20.8%), France (15.8%), the United Kingdom (9.7%), the Netherlands (8.9%), and Poland (7.7%) [22]. In 2019, 22.6 million cows were used for dairy in the EU-28 [22]. Germany had the largest dairy cow population with 4.0 million of cows, followed by France with 3.5 million, Poland with 2.2 million, Italy 1.8 million, Ireland 1.4 million, and 1.1 million cows in Romania [22]. In order to produce milk, each cow gives birth to one calf each year, but the technical coefficient often used is 0.9, to take losses into account. This means that around 22.6 million calves are born in the EU each year. About 35% of the calves are reared to replace slaughtered cows. Male calves and the remaining female calves (15 million calves a year) are redundant and have a very low financial value. Some dairy-type and cross-bred calves are reared for beef in intensive systems. Others, of a more extreme dairy type, may be destined for veal production or slaughtered as unwanted shortly after birth and before registration. Approximately 4.3 million heads of cattle were traded alive between EU countries in 2018 [23]. Belgium, Ireland, Greece, Spain, France, and Italy exchanged more than 1.8 million heads of cattle. The dairy beef production system and its limitations (recover at the entrance and reduce the risk of BRD) is also a challenge in other dairy production countries like USA or New Zealand. Britt et al. [24] analyzed the future of dairy farming in the next 50 years. These authors indicated that dairy beef will increase in importance because its production generates about one-third of the greenhouse emissions equivalents per unit weight of product compared with traditional beef production citing Opio et al. [25]. Cows with lower genomic ranks in herds will be inseminated with sex-selected semen from beef sires or will receive terminal cross embryos from beef-breed donors. This will increase the proportion of dairy farm income generated by the sale of animals, and these animals may enter a premium consumer market focused on climate-friendly beef products.

Therefore, as mentioned previously, these unweaned and unwanted calves are not a priority for the dairy sector and as a result their postnatal care has not been a priority for them. Until now, if they are considered a by-product or even a waste product, they have been treated as a by-product and in consequence they have received little attention and care. However, recently, different factors have provoked attention to these unweaned calves coming from the dairy farms: i) there is increasing public concern regarding the welfare of this long-distance transport of unweaned calves to the fattening farms [26], and this concern is under European Parliament supervision; ii) the non-rational antibiotic overuse at arrival to increase their survival that causes antimicrobial resistance; and iii) and the decline of beef cows population that reduces the beef calves in the markets [27]. Therefore, improving their health and welfare is a worldwide concern [28] and their increase in value may also help the dairy industry itself. 

The present review will first focus on nutrition before arrival at the rearing farm. Second, we will analyze the consequences and the impact of these feeding/management practices that take place before arrival on the immune system, and the digestive tract. Third, we will consider some potential nutritional strategies for future research.

## 3. Nutrition of Unweaned Calves’ Previous to the Arrival to the Rearing Farm

### 3.1. Nutrition before They Are Collected and Arrive to the Auction Market or Concentration Center 

One of the key aspects to develop a healthy gut is to ensure an adequate consumption of colostrum during the first hours after birth to provide immunity. Because of their type of placenta, the transfer of immunity from the dam to the offspring does not take place during gestation in ruminants. For this reason, newborn calves are born agammaglobulinemic [29] and their initial immune system development depends on the consumption of colostrum during the first hours after birth. Providing a good quality (IgG > 50 mg/mL) and quantity (5–6 L) of colostrum during the first 6–12 h after birth is key for the development of the immune function due to the high levels of maternal IgG contained in it [10]. Colostrum is also rich in bioactive proteins like IGF-I, IGF-II, growth hormone, insulin, prolactin and leptin [30,31], carbohydrates, and fatty acids [32], and for this reason it is considered as a natural prebiotic. Moreover, the development of the intestinal epithelia is compromised due to colostrum shortage [33,34]. The GIT plays an important role in the correct transfer of immunity by allowing the passive absorption of macromolecules (like Ig and lactoferrin) into the bloodstream via immature enterocytes [35]. In addition, the GIT constitutes a mechanical barrier separating the internal media from the external environment. Many specialized cells (like Goblet and Paneth cells, leukocytes, macrophages, M cells, and lymphocytes) are located in the GIT and their main function is to protect the intestine from external pathogens and to react to those generating immunological responses. It has been shown that the proliferation rates of epithelial cells increase with colostrum consumption when compared to milk replacer, increasing small intestinal villus height and the crypt depth of neonatal calves [34]. When colostrum is not properly provided, there is a failure of passive transfer (FPT) increasing the incidence of diarrhea [36], and if the adequate energy intake is not consumed, the risk of hypothermia rises, facilitated by the incapacity of newborn calves to efficiently thermoregulate [37]. Despite the general knowledge of the importance of feeding colostrum, a recent study based on surveys of calf management in dairy farms showed that some producers do not always feed enough colostrum to their male calves, they receive smaller amounts of colostrum, and/or the colostrum feeding time is delayed [28]. Renaud et al. [38], in a retrospective study, observed that 20% of the calves had a failure in the transfer of passive immunity and estimated that male calves had 0.14 g/L lower serum total proteins than female calves, indicating a worse passive immunity transfer. Even though there are different analytical methods to assess the failure of passive transfer in neonatal calves [39], nowadays there is no method to assess if colostrum intake was appropriate when calves arrive at the rearing farm at the age of 14 to 21 days. In Europe, as most dairy beef calves arrive to the rearing farm at the age of 14 to 28 days, this is an important gap that research should cover, and we should be able to find new biomarkers that could indicate that calves have been properly colostrated as this age is critical for their immunocompetence. Hulbert and Moisà [29] have written one of the best reviews that describes the stress, the immunity and management of calves. The mean age at which these dairy unweaned calves are transported coincides with the decrease in maternal antibody and the immaturity of the humoral immune system, so these animals are very vulnerable. These authors indicate that maternal antibodies from colostrum appear to stay in the calf’s system for the first 3 w of life. However, the calf’s cellular or humoral immune systems at 3 w of age do not appear to be vastly different from w 1 [40]. Through increased exposure to novel microbes, the calf starts to develop its own antibodies after about 3 w. 

Finally, and not less important than ensuring adequate colostrum feeding, is the adequacy of the feeding program of those calves before been sold and transported to the rearing farm or auction market. In a recent paper that summarized interviews with dairy farmers in England [41] it was highlighted that dairy farms often do not feed calves according to the recommended best practice despite the legislation and technical advisory efforts. Surprisingly, despite the huge amount of studies and reviews published in recent years that evaluate different calves’ feeding strategies on performance and health [42,43,44,45], these authors indicate that advisors were concerned because calves are commonly underfed, probably because there are insufficient clear recommendations. Consensus on calf feeding standards to ensure physiological function, facilitate weaning, support growth, health and welfare is another urgent research gap to improve nutritional status before they are sent to the auction market or concentration center.

In summary, several nutritional and management practices are critical before calves arrive to the rearing farms. These practices are: (i) improper colostrum intake; (ii) the average transport age of around 21 days of age makes the calf susceptible to infectious diseases because of their low immune protection; and (iii) that calves are underfed, predisposing them to have low energy storages that are mobilized under stress situations and needed as fuel for the immune response. In a recently published study, Marcato et al. [46] already indicate that before the arrival to the concentration center, calves already experience different sorts of stressors such as handling procedures, transport, as a well as different durations of feed and water withdrawal. Therefore, the calf starts its trip to the rearing farm not well prepared to overcome the stress situation that the auction market or the concentration center and transport generates.

### 3.2. Nutrition during the Transition Period: From the Farm where the Animal Is Born and Collected to the Final Rearing Farm (Transport—Auction Market or Concentration Center—Transport)

There is little information on the management and feeding practices during this transition period, like the description of the duration in days (hours) between the animals that are collected from the dairy farms, received and mixed in the auction markets or concentration centers, and transported to the rearing farm. Marquou et al. [47] described the calf’s health in five auction markets in the province of Quebec, producing around 80% of Canadian veal calves, however, important information was missing like the age, previous transport data, feeding program at the auction markets, or the duration of the stay at the auction market. Winder et al. [48] in a previous study in Canada observed that suppliers (auction markets) affected the calves’ mortality at arrival and remarked that access to more detailed data on source-farm calf management would likely be beneficial, although acquiring this information would be very labor intensive. Even though there is an increasing interest in the welfare and health of the dairy beef calves, there is scarce published surveys or data that describe the feeding programs of those calves at the auction markets or concentration centers, although there is some emerging literature (still insufficient) that describes it. For example, Wilson et al. [49] describe animal health and BW at North American auction markets and in their study they reported that calves only stayed one day in the auction market and they did not receive feed or water. Bernardini et al. [50], in a study with very few animals (14), followed their trip from a Polish dairy farm to the rearing facilities in Italy. These authors described that these animals were transported 70 to 90 km from the farms of origin to the transit center, where they were housed and remained for almost 30 h, and 3 h before the departure, the calves had ad libitum access to a hydration solution at a concentration of 20 g/L. Marcato et al. [46] described that in a German concentration center, the animals stayed one day and they studied the effect of feeding either once milk replacer or an electrolyte solution on the calves blood metabolites after arrival. During this transition period calves suffer an anorexia (fasting) which corresponds to the transport duration, but probably they may also suffer from undernutrition that corresponds to the time that they stay at the concentration center or the auction market, therefore the lack of this information is an important gap in order to improve their nutritional and immunological status, and therefore their health and welfare at arrival. 

## 4. Consequences of Previous Nutrition on Gut Health and Immune Status at Arrival to the Rearing Farm

The impact of calf transport on energy mobilization indicated by the increase in plasma concentrations of non-esterified fatty acids (NEFA) and beta-hydroxybutyrate (BHBA) and the decrease in glucose is well documented. Studies with unweaned calves confirm that this energy mobilization also takes place, and even their energy stores may be more limited compared with older weaned calves [26,49,51,52]. Information on calves’ characteristics has mostly been gathered from studies at the time of arrival in the rearing facility [48,53]. In few studies where blood samples were collected at the collection center or auction market before loading them to the rearing facility [46,50], it was confirmed that the calves already had a negative energy balance before being loaded. Knowles et al. [26] observed that feeding electrolytes and glucose in 8 h intervals during long journeys (over 16 h) reduced the energy mobilization of the calves. These authors suggested that this rest–stop feeding every 8 h (which implicates unloading and loading animals repeatedly) may be less effective than completing the transport journey as quickly as possible. Recent studies [46] confirmed that this energy mobilization may be modulated or reduced if animals were fed with milk replacer compared with electrolytes and if the transport duration (in consequence a fasting period) was shorter (6 h vs. 18 h). These preliminary data reinforce the fact that feeding strategies at the auction market or concentration center in combination with transport time need to be considered to improve the calf’s energy status minimizing their energy mobilization that increases during this transition period. These studies also indicate that the energy status is quickly recovered within the 24 h after arrival [26,46,50,53] as soon as the animal is fed. Marti et al. [53] observed that calves recovered NEFA and BHBA plasma concentrations at the farm arrival after they rested following a long transport journey (over 19 h) at a concentration center where the calves were rehydrated with electrolytes and dextrose and feed were provided. 

However, the question is, does this negative energy balance in young calves have a long-term effect on their nutritional and immunological status? When the calves arrived from auction markets or concentration centers at an age of 21 to 28 days, and were fed the amounts of milk required for their age compared with low-milk programs, the animals had mechanical diarrheas and there was an increase in morbidity [51]. Several studies conducted on bovines have demonstrated that the short-term feed restriction in beef cattle [52,54], or progressive feed restriction in dairy cows [55] or yearling beef animals [56] can affect tract barrier function. Furthermore, stress (physiological, prolonged exercise, and heat stress) has been shown to disrupt intestinal permeability, affecting normal digestive functions [57]. Wood et al. [58] observed a decrease in gut permeability during the weaning process which involves dietary changes (from liquid to solid feeding, from intestine to reticulo–rumen digestion, changes in energy source) being a stressful situation for the calves. Therefore, the previous life management of these dairy beef calves may have impaired gut barrier function. It is known that the increase in barrier function permeability due to feed restriction rises the risk of bacterial products like endotoxin translocation from the gut lumen into portal, lymph, and systemic circulation [59], exacerbating the inflammatory response. Dysfunctions of the intestinal barrier and alterations of the intestinal permeability are known as “Leaky Gut”. Endotoxin infiltration of the intestinal barrier activates the immune system and causes a well characterized inflammatory response: endotoxemia, pro-inflammatory cytokine release, local and systemic inflammation, possible multi-organ damage. The body mounts in an immune response which diverts energy and nutrients away from growth to the immune system as it will be discussed later. So, additional effects of fasting or feed restriction and stress of the calves before arrival may impair the intestinal permeability. Serum citrulline has been proposed as an in vivo biomarker for enterocyte mass, epithelial cell damage and absorptive function [59,60]. In patients with intestinal dysfunction (compromised intestinal function), citrulline levels decline compared with control patients [60]. Robles et al. [61] analyzed the serum of a subset (160) of total of 1601 calves (3–4 w of age; 65 ± 9 kg of BW) that were transported in a short journey (<9 h) or and long transport with a rest stop proving feed and water to the calves (>9 h). Serum citrulline concentration was lesser upon arrival in long transport (47.7 ± 3.39 μM) and short transport (42.6 ± 2.46 μM) compared with 2 w after arrival (55.9 ± 3.42 μM vs. 58.8 ± 2.52 μM for long transport and short transport, respectively). In a recent study with unweaned calves it was observed that the citrulline concentrations at the day of arrival and 2 days thereafter differed depending on feed restriction and anorexia severities and were less compared to control calves that were not feed restricted [54] (unpublished data). These data indicate that the feed restriction and anorexia may have long-term effects on gut functionality. Moreover, fasting and malnutrition have also been related to a reduction in the number of intestinal cells and villus height in addition to increments in cell apoptosis, decreasing the absorptive capacity of the gut [55,56]. If there is an increase in intestinal barrier disfunction, it may worsen the existing negative energy balance of the calves at arrival due to the reduction of digestion and the absorption of nutrients, and the redirection of nutrients caused by the activation of the immune system, decreasing the animal availability to cope with infectious diseases like BRD. In addition, lipopolysaccharide (LPS)-induced inflammation linked to gut permeability has an energetic cost that redirects nutrients from anabolic processes (growth) to inflammation that contributes to reduced glucose availability. Kvidera et al. [62] estimated that LPS-activated growing steers the amount of glucose utilized for the immune system was 1.0 g/kg BW0.75/h. These energetic requirements may aggravate the negative energy balance resulting from feed restriction, fasting and stress observed upon arrival after transportation. In addition, little is known regarding the effect of the negative energy balance increasing the susceptibility to suffer an infectious disease like BRD. In ewes, Bouvier-Muller et al. [63] using whole blood transcriptome analyses did not observed an interaction between negative energy balance and inflammatory challenge (mastitis). These authors described that blood leucocytes responded to a negative balance challenge, shutting down lipid-generating processes, activating fatty acid oxidation and inhibiting glucose oxidation and transport. The inflammatory status had an opposite response activating cholesterol and fatty acid synthesis and upregulating some glucose transport genes and increasing glucose utilization and oxidation. Blackebrough-Hall et al. [8] analyzed over 900 cattle looking for diagnostic biomarkers of BRD through serum metabolomics analyses. These authors observed that animals that were visually diagnosed as suffering BRD had increased serum hydroxybutyrate and decreased citrate concentrations and these authors indicated that this would be the consequence of the energy expenditure required for immune cell production with the onset of the diseased and increase glucose oxidation. These data support the hypothesis that calves suffering a feed restriction and fasting recover, 24 h after arrival serum glucose, NEFA, and BHBA serum concentrations, but this negative energy balance may have important long-term consequences on gut permeability and functionality.

Calves suffer also oxidative stress at arrival [64] which may modulate the intensity of inflammation in BRD after transport [64,65]. Oxidative stress is an imbalance between oxidants and antioxidants in favor of the oxidants, leading to a disruption of redox signaling and control and/or molecular damage [66]. The generation of pro-oxidant chemical species is one of the most evident consequences of inflammation among other causes like injury, stress or physical exercise. During innate response, cells like neutrophiles generate reactive oxygen species (ROS) and antioxidant systems guarantees the proper functioning of T cells and there is a controlled immune response [67]. The excessive/prolonged release of superoxide is potentially co-responsible for enhancing systemic inflammation and oxidative stress [67]. Free radicals and ROS and reactive nitrogen species (RNS) are capable of damaging all types of biologically relevant molecules (DNA, proteins, lipids and carbohydrates), mainly lipid peroxidation and protein oxidation damaging the cells. Therefore, traditionally these ROS and RNS compounds have been viewed as toxic compounds and an imbalanced antioxidant level should be avoided. Immune cells are particularly susceptible to oxidative damage as they generate large amounts of ROS and have a high percentage of polyunsaturated fatty acids (PUFA) in their membranes which makes them more susceptible [68]. Therefore, further studies need to elucidate in those calves that arrive with a negative energy balance an oxidative stress and with a compromised barrier function, all processes concomitant affecting the immune system and the energy requirements and altering the animal’s susceptibility to an infectious disease like BRD. 

Moreover, animals arrive starving, and they may eat fast and large amounts of feed at arrival without having a digestive tract prepared to digest them properly as described above; in consequence, they have indigestion and a reduced intake having difficulties to achieve the desired consumption. In weaned calves, it has been observed that not all calves eat during the first 2 w after arrival [69]. When transported long distances to begin eating, more than 4 d may be required for all healthy calves, or even more time may be required for morbid calves. Sometimes, animals are also tired, and it takes time until they visit the feeder. This study with weaned calves [69] highlights the importance of the previous experience of calves with the different feeding programs and feeders. Calves unfamiliar with eating from a bunk only consumed 0.5% of their body weight during the first 7 d compared with the calves familiar with the diet and feed bunk that consumed 1.5% of their BW. In unweaned calves, the impact of a previous feeding program (restriction) and transport hours (fasting time) also affects the feed intake pattern upon arrival. Pisoni et al. [70] evaluated in unweaned Angus–Holstein bull calves the effects of different degrees of feed restriction and anorexia (simulated by the auction market and transport) on the concentrate intake and BW recovery. The BW was greater for control calves (not submitted to feed restriction and anorexia) compared with the rest from d0 to 7, while the BW of calves submitted to severe feed restriction (hydrate solution and 19 h of feed withdrawal) was lesser compared with the rest from d−1 to d7. On the day after arrival, the concentrate intake did not differ among treatments, however, the second day after arrival, the concentrate intake of those calves that suffered feed restriction and fasting independently of the degree (mild, moderate and severe) was lesser and increased throughout the first week, although severe feed-restricted calves started to recover the concentrate intake at day 5 after arrival (Figure 1). Therefore, the previous nutrition (restriction and fasting) and previous type of liquid feeding (MR or HS) offered affects concentrate intake recovery. To achieve the maximum concentrate intake after arrival is important to stimulate rumen development and to maximize energy intake; in the case that we do not succeed in maximizing concentrate intake throughout the first week of arrival, the negative energy balance at arrival will worsen and we will not be able to meet the maximum growth requirements when feed intakes are low.

In addition, calves transported in a range from 4 to 20 h arrived at the rearing facilities suffer from anemia (indicated by low hemoglobin and hematocrit values) in addition to the negative energy balance and an inflammatory process [53]. Steinhardt and Thielscher [71] also observed low hemoglobin values (blood hemoglobin less than 6.0 g/L) in veal calves at the farm arrival. Iron deficiency anemia is caused by an undersupply of iron during the exclusive milk feeding (veal production), chronic blood loss (gastric ulcers), infestation with parasites or malnutrition [72]. As a consequence of their anemic status, the health and robustness of calves are affected and are more susceptible to diseases, lack of appetite, and compromised ability to cope with physical stress [73,74]. Iron deficiency affects humoral and cell-mediated immunity [75]. Therefore, in the white veal calf’s industry where the calves are not fed solid feeds, the anemia is monitored [74]. Outside this white veal calf’s industry, it is surprising that the little attention has been driven on the incidence and the duration of these anemia and the possible impact that this anemia may have on animal health and recovery. However, the determination of serum ferritin would be necessary to confirm the diagnose that these calves suffer from iron deficiency anemia [72].

## 5. Considerations to Advance in Future Nutritional Strategies

After this first analyses, some important gaps have been detected that need to be studied in order to design the future nutritional strategies to reduce the susceptibility related with nutrition to the BRD of unweaned calves at arrival. Both an effective gut and an effective immune status are linked [63] and it is crucial to recover the animal’s gut and immune status [76] as soon as possible and to overcome possible BRD episodes.

### 5.1. Considerations for Research to Prevent through Nutritional Strategies BRD

Some considerations have been mentioned previously as the need to find a method to assess if colostrum intake was appropriate when calves arrive at the rearing farm at the age of 14 to 21 days, which is also an important research gap that need to be fulfill in the following years.

Additionally, we should prevent as much as possible feed restriction and fasting that causes gut permeability and a mild systemic inflammatory status. Before designing any strategy we should take into account more detailed information about the feeding programs implemented and transport characteristics (duration, feeding) from the dairy farm, the auction markets and concentration centers, and transports to the rearing farm. Nowadays, from the scarce information that we have we know that one of the most common reported feeding practices in auction markets or concentration centers is the supplementation of rehydration solutions. Reviewing rehydration there is a great variability in protocols and most strategies in calves are based on oral electrolyte solutions designed for calves with diarrhea and aim to restore hydration and to correct metabolic acidosis [77,78]. The amount of glucose added, if any, is limited and is added to facilitate sodium absorption and to provide an energy source for the calf but the requirements of energy and protein are not achieved compared to whole milk or milk replacer (MR). For example, is a rehydration solution better than feeding milk replacer? Marcato et al. [46] recently evaluated the effect of pre-transport diet on a calf’s recovery based on their physiological status (blood metabolites, hemogram, BW, rectal temperature and skin elasticity). These authors [46] observed that feeding milk replacer (1.5 L of a mixture of 125 g of milk powder in 1 L) had a positive impact compared with feeding electrolytes (1.5 L of a mixture of 20 g of electrolytes in 1 L of water) only when transport duration was 8 h. In longer trips (18 h), the pre-transport diet did not have an effect on animal recovery. This is one of the published studies that evaluates different pre-arrival (pre-transport) diets on animal recovery. However, there are many open questions to be solved. Should the rehydration solution or the MR be designed to avoid the fasting/feed restriction effect on intestinal health? To prevent animals being hungry, and feed restricted in the auction markers, strategies like offering them ad libitum concentrate and hay and/or acidified milk be implemented? In addition to continuous energy provision, solid feed (concentrate and straw) supplementation could additionally stimulate the rumen function which could provide them a continuous energy supply (even for the trip) and prepare them better to consume solid feeds at arrival. However, how should these concentrates be designed (formula, presentation form) to stimulate intake and be easily managed? Another option could be to feed in the auction market or concentration center acidified milk ad libitum additionally to the milk replacer or rehydration solution. Acidified milk can be available in the pens for many hours as the addition of organic acids helps control bacterial growth. In studies with replacement heifers (not dairy bulls coming from auction markets) where acidified milk was provided with similar composition and amounts of a standard milk replacer, no differences in the average daily gain were observed [79,80,81]. However, when acidified milk is offered ad libitum, it may reduce the incidence of disease by allowing the calves to consume more nutrients from the milk [81]. When feeding acidified milk, one should have in mind that one possible disadvantage of the acidification of the milk is that it alters the taste and reduces palatability [82]. One of the main difficulties in the pre-transport diet is that calves usually stay between 1 and 3 days in the auction market or concentration center, so there is little time for nutritional intervention. Moreover, the impact of last feeding before being loaded (time lapse before loading, nutritional quantity and quality) and the interaction with transport duration are factors that should be also studied as they will have a great impact on the fasting consequences on digestive and immune status.

In addition, before thinking up any strategy, we should characterize the feed restriction (type of nutrients, feed presentation, duration) and anorexia (duration, additive effects with other stressors) and its consequences on the digestive and immune metabolism and status. In rats, Habold et al. [83] described the metabolic changes in glucose intestinal metabolism and its relationship with different fasting phases. These authors [83] cited that fasting induces an increase in gluconeogenesis in the small intestine which produces up to one-third of endogenous glucose after 72 h of fasting [84]. They described different fasting phases, in phase I, where there is a rapid depletion of glycogen, during phase II, lipid stores are progressively oxidized whereas body proteins are efficiently spared and intestinal cells increase their gluconeogenesis using glycerol as fuel. In a later fasting phase (phase III), characterized by both a strong increase in protein utilization as a substitute fuel for lipids and a rise in plasma corticosterone level, the fuel for the intestinal cells to synthesize glucose are amino acids (mainly glutamine). Habold et al. [83] also indicated that during dietary restriction, the affinity of the carriers for sugars increases [85] and a 72 h fast causes an overall increase in glucose absorption in rats [86] and the increased epithelial permeability due to fasting may induce the paracellular movement of glucose. In which the fasting phase are our animals at arrival? Which are the factors that modify the susceptibility of an animal to go from a fasting phase to another? Should we feed them differentially depending on the fasting phase? In unweaned calves, do gut (intestinal) cells modify their metabolism during and after fasting? However, does glucose transport or glucose metabolism of the intestinal cells change during fasting or feed restriction? Could these changes be important in order to design feeding strategies to recover the gut health of those calves? As mentioned previously, during fasting, the glucose sources and gluconeogenic precursors vary in relation to changes in the utilization of body fuels. So, coming research should help us to understand better the intestinal nutritional requirements under these stressful and feed restricted and anorexia situations. 

### 5.2. Considerations for Research to Recover Animal Gut and Immune System Functionality through Nutritional Strategies to Prevent BRD at Arrival to the Rearing Farm

Different feeding strategies that could be easily implemented and their impact on animal health and recovery at the farm should be evaluated. The main goal of those strategies should be that the animals recover from dehydration, hypoglycemia and oxidative stress and restore the gut barrier health altered during transportation (which impacts the immune system and intestinal health), and to prevent their negative energy status worsening. Here, we will focus on the recovery of gut restoration, gut maintenance, gut oxidative stress and to recover animals from dehydration and hypoglycemia. 

Focusing on gut restoration, the current nutritional strategies upon arrival are focused mainly on gut health maintenance, but the first step is the restoration of barrier function. The restoration of gut health has been intensively studied in cultured cells [87]. Blisklager et al. [87] described numerous growth factors and cytokines involved in the epithelial restitution and most of them are conveyed in the expression of transforming growth factors (TGF)-beta-dependent pathway. Other compounds involved in gut healing are polyamines (putrescine, spermidine, spermine) and trefoil peptides. In the closure of the paracellular space, there is accumulated evidence that prostaglandins have a role in the tight junctions’ recruitment and repair, and that those mechanisms are related to their ability to modify ion transport. As mentioned previously, those unweaned calves could be suffering from a “leaky gut” syndrome that has a high risk of causing a LPS challenge (sepsis), to understand that the restoration mechanisms of the gut barrier in those unweaned calves should be a priority. Moreover, Wang et al. [88] in mice observed that specific metabolic programs are coupled to different types of inflammation to regulate tolerance to inflammatory damage. Furthermore, glucose supplementation (or deprivation) may have opposite effects depending on type of inflammation (bacterial or viral). These authors indicated that there is evidence that the fasting response that occurs as part of the inflammatory response in an LPS sepsis is required to maintain resistance to oxidative stress limiting the ROS induced by anti-bacterial inflammation. Their work indicated that the timing of ketogenesis, an adequately nourished host, or both were necessary for the protective effect of fasting that occurs as a coordinated response to bacterial inflammation. This work highlights that energy sources supplemented when an animal suffers from a leaky gut should be considered, as to our knowledge no work evaluating different energy sources has been done considering the possibility that most of the unweaned calves that arrive are suffering from leaky gut. Thus, first, we should be able to restore and repair and then refeed the gut. Thinking of restoring feeding transition milk (two to six milkings after calving) like colostrum contains great amounts of bioactive compounds [89]. The milking number affects a great amount of milk proteins; the majority of the 78 proteins are encompassed within catalytic activity, binding, cellular process. Therefore, rethinking the opportunity of feeding transition milk to calves at arrival and evaluating its impact on gut restoration rather than discarding this transition milk seems to be a good opportunity that needs to be explored as recently proposed by Pyo et al. [90]. 

As mentioned previously in recent years, most nutritional strategies at arrival have been focused to maintain and modulate the gut health [91] and can be split into three strategies: (i) the maintenance of a stable and beneficial microflora that prevents pathogen colonization, increases digestive capacity, lowers pH, produces beneficial metabolites like SCFAs, enhances barrier integrity and improves mucosal immunity; (ii) maintenance of a healthy and functional gut barrier which means maintaining the epithelial cells and mucus layer (mucin glycoproteins secreted by Globet cells); (iii) maintenance of effective immunity as the gut possesses the largest mass of lymphoid tissue in the body (GALT: gut-associated lymphoid tissue). Most nutritional strategies are based on ingredients or additives that have been more widely studied in monogastrics than in ruminants. Most additives tested are probiotics, prebiotics, metabolites (butyrate as energy source for endothelial cells), short-chain fatty acids, phytogenic feed additives, minerals, vitamins, antimicrobial peptides, etc. Duff and Galyean [5] wrote an excellent review focused on the management of stressed newly arrived feedlot cattle, however, most references or cited studies were based on weaned calves that had a fully developed rumen. There is a great opportunity in learning from all nutrition knowledge generated in recent years with replacement heifers around stressful weaning [43]. When analyzing the published literature of the different nutritional strategies studied in replacement heifers’ that could be inspiring for future strategies to be implemented in dairy beef cattle, one realizes that there that there is an increasing number of studies that measures their impact on gut permeability or epithelium structure. These parameters are fundamental to choose if these strategies are potential strategies to be implemented in dairy beef cattle. For example, Wilms et al. [92] observed that substituting lactose with dextrose in the milk replacer at 3 and 7 weeks of age increased intestinal permeability. These authors argued that this hypertonic milk replacer which caused intestinal osmotic pressure was greater than plasma osmolality and that this could exert an osmotic pressure on cell lining and on the tight junctions. Amado et al. [93] studied feeding in male calves high-fat milk replacers, and observed an increased intestinal permeability compared with lactose-rich milk replacers. Curiously, Urie et al. [94] observed a significant association between the amount of fat per day in liquid and increased calves’ mortality; so, the effect of fat content, fat type in the milk replacer and the total amount of fat consumed on gut permeability should be further studied. Pisoni and Relling [95] hypothesized that supplementation with a prebiotic (yeast fermentation products) could stimulate the secretion of the gut hormone glucagon-like peptide 2 (GLP-2) by inducing changes in gut microbiota. GLP-2 is associated with intestinal permeability regulation and could enhance the mitogenic action of IGF-1 to synergistically benefit GIT development [93]. However, these authors [95] did not observe benefits of prebiotic supplementation on performance parameters, plasma GLP-2 concentration, intestinal permeability when supplementing these yeast fermentation products in the milk and starter in heifers. Strategies based on maintaining gut health and through nutritional strategies like additive supplementation are mainly preventive strategies and it is difficult to observe an improvement in performance. Finally, little attention has been paid to potential harmful ingredients that could impair gut barrier function like ingredients rich in mycotoxins. Aflatoxins, ochratoxin, and deoxynivalenol have been shown to impair intestinal permeability in different species, and recently, the effect of mycotoxins on Goblet cells and mucin production, and on the enteric nervous system and microbiota has been evaluated [96]. 

When focusing on the recovery from oxidative stress, an animal’s antioxidant defense mechanisms are based on the synthesis of antioxidant enzymes, glutathione, thioredoxin and coenzyme Q. In animal nutrition, traditionally, to help in the antioxidant defense mechanism in preventing this imbalance, different antioxidants have been supplemented. The most common antioxidant compounds supplemented in animal nutrition are vitamin E, carotenoids, minerals (selenium), and secondary plant metabolites. For example, dietary polyphenols and their metabolites exert beneficial effects through a combination of mechanisms that include a reduction of the inflammation and the oxidative stress. Important is that their bioactivity depends on the direct reaction with the oxidized species, so it depends on their solubility, absorption, distribution, and metabolism. The incubation of leukocytes collected from animals that have been transported with alpha-tocopherol or ascorbic acid reduced the peroxidation intensity and increased the antioxidant capacity and leukocyte viability [65]. However, these data observed in vitro are not always observed in vivo [68,97]. Unfortunately, an animal’s shelf-life cannot not be prolonged like food shelf-life. The antioxidant treatments have little or no impact on animal ageing, for the reason that antioxidant protection is an evolved strategy which is billions of years old that resists being easily tampered [66], or as mentioned previously in many cases, the bioavailability of these antioxidants may be limiting. Traditionally, these ROS and RNS compounds have been viewed as toxic compounds, however, recently their role as cell signaling pathways has been recognized [98]; therefore, maybe they should not be modulated in order to guarantee a proper immune response. In addition, as [98] indicates, there is no single test that indicates the optimal oxidative status of an animal: the oxidative status index (like the ferric-reducing ability of plasma) together with macromolecule damage (lipid or protein oxidations products like thiobarbituric acid reactive substances or leukocyte viability) should be combined. Therefore, the scientific community needs to further study the role and consequences of the oxidative stress after transport, which is the threshold of the antioxidant status that would increase the BRD incidence symptomology and standardize a methodology to estimate it. In this sense, studies that analyze the whole-blood transcriptome data of blood leucocyte of animals submitted to feed restriction and inflammation [63] or that suffer BRD [98] are crucial in the future to understand the metabolic processes and design future therapies alternative to antimicrobial use. Scott et al. [99], based on their results, suggest that vitamin E supplementation together with aspirin may be good treatment for BRD as their metabolites induce specialized pro-resolvin mediators (SPMs) expression, signaling molecules that are produced by leucocytes and macrophages derived from PUFA metabolism that may protect calves from BRD and prevent from chronic inflammation.

Finally, when focusing on rehydration and feed intake recovery, as mentioned previously, a rehydration solution could be administered upon arrival, but this solution should be rethought for animals that have suffered anorexia and stress. Additionally, water and feed intake should be encouraged from the very beginning after arrival to recover calf hydration and glycemia. To recover feed (concentrate) intake as soon as possible, we should think beyond the traditional nutritional strategies (ingredients, additives that affect gut and immune status, aromas, feed presentation), the factors related to feed management may be very limiting and should be carefully reviewed. For example: feeder design, pen density (competition at the feeder), preconditioning, water temperature and availability, vaccination programs and prompt disease diagnosis could temporarily decrease feed intake. Although this paragraph is very short, the amount of research to be done in this area is very large. 

### 5.3. Other Important Considerations for Research Related with the Nutritional Strategies to Prevent BRD

This section summarizes a group of miscellanea considerations that also need to be considered. First, all the studies evaluating the aforementioned gaps need a golden standard, a control group that does not suffer transport, mixing, feed restriction, fasting, etc., in order to properly evaluate the impact of each intervention. This golden standard is also very important, as it is the reference of an ideal situation where no animal movement would take place, if we can design an intervention who’s impact is close to the golden standard, we can guarantee that this production practice has no impact on animal welfare and does not increase the antibiotic use, as we will be able to accomplish with the societal demand.

Second, the research outputs should also provide information to design risk categories in order to classified calves into suitable or unsuitable to be transported based on their physiological status and/or based on their previous life quality (before marketing, transport, and marketing), or future transport characteristics (transport duration, season). This screening process should aid in reducing animals that cannot properly recover at arrival and therefore their welfare is impaired and decreases the severity of BRD.

Third, the importance of age or BW in the success of possible nutritional strategies should also be considered. It has been observed that a greater weight upon arrival is protective for early mortality [48,100]. Weight at arrival is related with breed, age, previous nutrition, dehydration, defection and urination during transport. It is unclear which of the variables that affects BW contributes more to BW variation and in consequence, the study of the mechanisms whereby BW affects mortality is difficult. Focusing on designing future nutritional strategies, one important question is the relationship between BW and the degree of digestive tract maturity. We know that during the weaning period, large physiological changes in the digestive tract gut take place, mainly as solid intake takes place the rumen starts to develop, but there are also microbiological and molecular changes in the lower gut [44], and in the enzymatic specific activities like pancreatic enzymes [101]. Most changes are related to solid feed intake and dietary composition, and again it is difficult to split the impact of age from BW on those changes. Nutritional strategies at arrival should be designed whilst keeping in mind the gut maturity and that these GIT changes take place within few weeks and are affected by previous nutrition. Usually, upon arrival in the rearing farm, we group animal by BW and within a pen or group, animals may differ between 2 to 3 w of age. Keeping in mind that BW may not be a good predictor of gut maturity as it is affected by transportation and that age may also not be related to gut maturity, should animals be grouped by age rather than BW or should we create new criteria for grouping arrival which takes into account both BW and age? More research is needed on these unweaned dairy beef calves that are transported to better understand the degree of gut maturity of the calves upon arrival and which are the most influencing factors to be able to properly design the nutritional strategies at arrival. Moreover, these new classification criteria at arrival could also help us in a screening process at arrival. For example, old animals with a small BW may indicate that they have been suffering from a disease or have not been properly fed previously; in consequence, this animal should be housed separately (possible focus of infections for the other animals). Although as indicated before, the best option would be that this animal would have been rejected in a screening procedure before being loaded and would not arrive to the rearing farm.

Fourth, nutrition management may help as described before to recover the animal as soon as possible and reduce the BRD incidence, but we should not forget other critical factors in the BRD onset like: (i) improving the immunity—increasing a significant proportion of the population that has immunity (e.g., vaccines type and vaccination timing, natural acquired or colostrum acquired immunity) hindering diseases to spread between individuals if a proportion is already immune, breaking the ‘chain of infection’; (ii) reducing stress factors that promote immunosuppression (e.g., weaning, mixing); (iii) reducing animal husbandry practices that promote contagious diseases (e.g., stocking density, transport of animals, comingling animals from different sources, poor biosecurity); and (iv) characterizing the pathogens involved in the BRD of the farm and their antibiotic resistance profile.

Fifth, a community of practice approach is needed to solve this problem. Unweaned dairy calves welfare is among the urgent issues that needs to be faced in the following years and requires leveraging capabilities across the scientific community and also across all players (dairy farmers, transport companies, auction markets, concentration centers, rearing farms, veterinarians, nutrition companies, policy makers…). It needs a community of practice approach. Communities of practices are defined as groups of people who share a concern for something they do and learn how to do it better as they interact regularly. Without this approach, without building the bridges between all players and integrating them in a community of practice we will not the able to go one step further and improve the welfare of these unweaned dairy beef calves. The first step, recognizing that there is problem, is currently happening. The second step to successfully break the barriers and build those communities with the most relevant players, debate and define common objectives, and design the research plans necessary to generate the necessary knowledge to convert the current problem into a past problem.

## 6. Conclusions

Improving the health and welfare of unweaned dairy beef calves is a worldwide concern. Reviewing the nutrition and management of these unweaned calves before arrival to the rearing farm and possible consequences like the negative energy balance, the increased permeability, the oxidative stress, the anemia, and the difficulties in feed intake recovery that been discussed. Several research gaps have been detected and the future nutritional interventions should be focused on the prevention of the negative consequences of previous nutrition and on the recovery of the gut and immune status. This review highlights that there is room for improvement and that there are new opportunities in designing nutritional strategies that could reduce the incidence of BRD and the antimicrobial use of unweaned calves upon arrival should be deeper studied.

## Figures and Tables

**Figure 1 animals-10-01908-f001:**
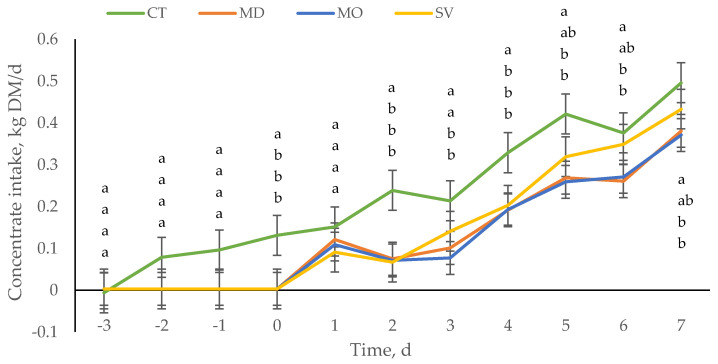
Concentrate intake (kg DM/d) of unweaned Angus–Holstein bull calves according to the control (CT): calves fed from d−4 to −1, 2.5 L of milk replacer (MR) twice daily, concentrate and straw ad libitum; mild (MD): calves fed only MR (d−4 to −1) as described for CT, and at the end of d-1 feed withdrawal for 9 h; moderate (MO): calves fed only MR as described for CT and at the end of d-1 feed withdrawal for 19 h; and severe (SV): only 2.5 L of a hydrate solution (HS) twice daily (d-4 to -1) and at the end of d-1 feed withdrawal for 19 h. After d 0, all calves received feed, water and straw ad libitum [54]. When letters differ in the same time point (day), it means that there are statistical differences between treatments.

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
