# Peer review of "Strategies for Feeding Unweaned Dairy Beef Cattle to Improve Their Health"

_animals, 2020, doi:10.3390/ani10101908_

Round 1
Reviewer 1 Report
The manuscript aims to review the breakthroughs in feeding of dairy calves that are used in feedlots for meat production. Although the manuscript provides substantial new information the way it is presented is confusing and repetitive and very difficult to follow. It has a very extensive introduction where it fails to set the ground for the following sections and is not explicit of its major objectives. There is a lot of reference to postnatal care and many repetitions. The author must shorted the introduction and make it more focused on the issues that will be later discussed. The issue of BRD is mentioned in several parts of the manuscript without a concise description of its relevance to the nutritional management. When describing the "breakthrough" and the "future nutritional strategies" there are several references cited with most of them being scattered in the text without a coherent manner of discussion. Overall, the manuscript has some important information that merit publication but there is need for rearrangement of the sections, reducing the citations and focus on the important issues that will be explicitly stated in the objectives.
Author Response
Changes in the paper are “track changes”, comments/answers or changes are written in italics after each reviewer’s comment.
Authors' answers are in italics below each comment
Reviewer 1
The manuscript aims to review the breakthroughs in feeding of dairy calves that are used in feedlots for meat production. Although the manuscript provides substantial new information the way it is presented is confusing and repetitive and very difficult to follow. It has a very extensive introduction where it fails to set the ground for the following sections and is not explicit of its major objectives.
After analyzing all reviewers comments the structure of the paper has been changed, more sections have been introduced and the objectives have been rewritten (line 152-155 of the current submitted version).
There is a lot of reference to postnatal care and many repetitions.
In the last years new studies in this topic have been published, therefore in this section there is more literature references. Repetitions have been tried to avoid.
The author must shorted the introduction and make it more focused on the issues that will be later discussed.
Has been spliced in two parts.
The issue of BRD is mentioned in several parts of the manuscript without a concise description of its relevance to the nutritional management.
The BRD description has been moved to the beginning (line 61 to 64)
When describing the "breakthrough" and the "future nutritional strategies" there are several references cited with most of them being scattered in the text without a coherent manner of discussion.
Changed following reviewer’s considerations.
Overall, the manuscript has some important information that merit publication but there is need for rearrangement of the sections, reducing the citations and focus on the important issues that will be explicitly stated in the objectives.
Regarding the structure and objectives, see previous comments, regarding the citations in a review are always large, we only gave the most relevant
Reviewer 2 Report
This is a well planned review with valuable information and gaps for future research identified. Considering the influence in calves welfare the implications of feeding management could be relevant.
There is some formal mistakes that should be changed:
- self-citationsin at some points as Lines 273, 310, 378,
- line 49 Edward 1996 and Pardon et al 2012 are not numbered
- lines 338 to 154 in a single paragraph are too long, I strongly recommend to divide them into 3 to 4 paragraphs.
- line 101 remove van Engen and Coetzee from the text as 20 reference it is include.
- lines 232 to 237 with different front tipe.
- line 262 "Recent studies (49) studies ..." delete the second studies
- line 280 Pisoni et al 2020 unpublished not reflected in References.
- line 282 phycological I guess it is physiological
- lines 305 and 306 not clear, please re-write.
- line 327 "capacity of the when the..." not appropiate expression.
- Line 418 "The achievement the maximum concentrate..." not appropiate expression.
- Lines 450 to 451 "These authors cited (81) that described that fasting..." and Line 457 "These authors (80)..." could lead to some misunderstandings.
- Line 613"whole blood transcriptome blood..." not appropiate expression.
Author Response
Authors' answers are in italics below each comment
This is a well planned review with valuable information and gaps for future research identified. Considering the influence in calves welfare the implications of feeding management could be relevant.
There is some formal mistakes that should be changed:
- self-citations in at some points as Lines 273, 310, 378,
Reworded to avoid self-citations
- line 49 Edward 1996 and Pardon et al 2012 are not numbered
Corrected
- lines 338 to 154 in a single paragraph are too long, I strongly recommend to divide them into 3 to 4 paragraphs.
Changed
- line 101 remove van Engen and Coetzee from the text as 20 reference it is include.
Done
lines 232 to 237 with different front tipe.
Changed
- line 262 "Recent studies (49) studies ..." delete the second studies
Done
- line 280 Pisoni et al 2020 unpublished not reflected in References.
Added as under preparation
- line 282 phycological I guess it is physiological
Corrected
- lines 305 and 306 not clear, please re-write.
Rephrased
- line 327 "capacity of the when the..." not appropiate expression.
Rephrased
- Line 418 "The achievement the maximum concentrate..." not appropiate expression.
Rephrased
- Lines 450 to 451 "These authors cited (81) that described that fasting..." and Line 457 "These authors (80)..." could lead to some misunderstandings.
Rephrased
- Line 613"whole blood transcriptome blood..." not appropiate expression.
Rephrased
Reviewer 3 Report
The manuscript entitled Breakthroughs for Feeding Unweaned Dairy Beef Cattle to Improve their Health from Devant and Marti is an important review since deals with a poorly studied situation, the fatting as beef or unwanted dairy calves. The authors fall into my criteria of the requirements to write a review paper, they have extensive published work in the field.
One personal note, the problem deal within the manuscript it´s like “cry over spilled milk”. The best strategy would be using sexed semen to produce female pure dairy heifers and male F1 dairy x beef crosses.
The expression Dairy Beef cattle is not quite common, it should be explained in the abstract, not only in the simple summary. Additionally, the introduction mentions mostly beef animals, so I am a little bit lost with the focus of the paper. BRD is a main concern in beef cattle raised at tie stall. How is the breeding system for this dairy beef cattle? Which are the main diseases? This could be placed in more detail in the manuscript. In my country, such animals are raised extensively (grass-feding), thus BRD is not a problem. Ticks and parasites are the most important diseases for such animals. Brazil such The introduction is too long. Must be shortened and a new topic/subtopic should be included to explain what the dairy beef cattle are, how they are raised, what are it main diseases, and their economical importance. I think this change will improve the readability and understanding of the manuscript.
I am not sure if the word Breakthroughs is adequate. Breakthroughs mean something already discovered, a Breakthrough discover that will revolutionize the field, which was not the case in this review.
The title must be changed to something more adequate with the text: Strategies for feeding Unweaned Dairy Beef Cattle to Improve their Health, or maybe: Strategies Improve Unweaned Dairy Beef Cattle Health through adequate feeding.
Introduction, L49: l [Edward, 1996; Pardon et al., 2012]
Please correct the citation format.
Figure 1. I am not sure if it´s adequate to use a previously published figure (if it was the case).
Regarding item 4 of the manuscript, it´s quite disorganized. Some topics are just an idea:
4.1.1 We need to find a method to assess if colostrum intake was appropriate when calves arrive at the rearing farm at the age of 14 to 21 days, this is an important research gap that needs to be fulfilled in the following years.
4.1.2. We should prevent as much as possible feed restriction and fasting that causes gut permeability and mild systemic inflammatory status. However, before designing any strategy we should:
4.1.2.1. Have more detailed information about the feeding programs implemented and transport characteristics (duration, feeding) from the dairy farm, the auction markets, and concentration centers, transports) to the arrival farm.
Without any citation whatsoever (not ok for a review manuscript).
And other topics in this same topic 4 has a detailed literature review.
I believe that this topic 4 must be corrected to include only the parts with proper literature review and comments, and not placed as breakthrough or ideas. They shall leave the suggestions for the conclusions, which could be changed to final considerations and then, list some of the author's ideas of what it needs to be discovered. Doing this, the authors separate what is proved science and what is pure speculation.
Author Response
The manuscript entitled Breakthroughs for Feeding Unweaned Dairy Beef Cattle to Improve their Health from Devant and Marti is an important review since deals with a poorly studied situation, the fatting as beef or unwanted dairy calves. The authors fall into my criteria of the requirements to write a review paper, they have extensive published work in the field.
- One personal note, the problem deal within the manuscript it´s like “cry over spilled milk”. The best strategy would be using sexed semen to produce female pure dairy heifers and male F1 dairy x beef crosses.
Dear reviewer, having male F1 crossed is a very nice way to increase the value of these calves. However, the problem would still persist, first because the success of sexed semen is not 100% and number of the replacement heifers needed are around 40%, so we still have these 60% of calves that are not desired and that are considered a waste product, even if they are crossbred animals and are better payed, usually they are fed improperly during the first days of life and transported.
- The expression Dairy Beef cattle is not quite common, it should be explained in the abstract, not only in the simple summary. Additionally, the introduction mentions mostly beef animals, so I am a little bit lost with the focus of the paper.
Thanks for the comment, the reviewer is right, this expression is not common, but really describes that cattle are from dairy origin, and we want to emphasize it as we think this would be the next 10-year topic in ruminant research (see below- different research centers web links that use this term). After another reviewer and yourself has indicated us that the introduction was too long, we will split it 2 parts and better explain the concept Dairy Beef cattle as you suggest below, we hope this helps, thanks for mentioning it, I think it will help to improve the paper.
Dairy calf to beef-https://www.teagasc.ie/animals/beef/dairy-calf-to-beef/
Dairy beef production-https://extension.psu.edu/dairy-beef-production
Dairy beef-https://www.dairyaustralia.com.au/farm/animal-management/animal-welfare/dairy-beef
- BRD is a main concern in beef cattle raised at tie stall. How is the breeding system for this dairy beef cattle? Which are the main diseases? This could be placed in more detail in the manuscript. In my country, such animals are raised extensively (grass-feding), thus BRD is not a problem. Ticks and parasites are the most important diseases for such animals. Brazil such
References of the importance of BRD (economic and morbidity) are given in the paper
- The introduction is too long. Must be shortened and a new topic/subtopic should be included to explain what the dairy beef cattle are, how they are raised, what are it main diseases, and their economical importance. I think this change will improve the readability and understanding of the manuscript.
See comment above.
- I am not sure if the word Breakthroughs is adequate. Breakthroughs mean something already discovered, a Breakthrough discover that will revolutionize the field, which was not the case in this review.
The title must be changed to something more adequate with the text: Strategies for feeding Unweaned Dairy Beef Cattle to Improve their Health, or maybe: Strategies Improve Unweaned Dairy Beef Cattle Health through adequate feeding.
Changed after reconsideration
- Introduction, L49: l [Edward, 1996; Pardon et al., 2012]
Done
- Figure 1. I am not sure if it´s adequate to use a previously published figure (if it was the case).
The figure has not been previously published, the data have been previously published. This figure is very important to illustrate the impact of feed restriction of those calves at feed intake at arrival, is the key of the problem.
- Regarding item 4 of the manuscript, it´s quite disorganized. Some topics are just an idea:
4.1.1 We need to find a method to assess if colostrum intake was appropriate when calves arrive at the rearing farm at the age of 14 to 21 days, this is an important research gap that needs to be fulfilled in the following years.
4.1.2. We should prevent as much as possible feed restriction and fasting that causes gut permeability and mild systemic inflammatory status. However, before designing any strategy we should:
4.1.2.1. Have more detailed information about the feeding programs implemented and transport characteristics (duration, feeding) from the dairy farm, the auction markets, and concentration centers, transports) to the arrival farm.
Without any citation whatsoever (not ok for a review manuscript).
And other topics in this same topic 4 has a detailed literature review.
Authors' answers are in italics below each reviewer's comment
I believe that this topic 4 must be corrected to include only the parts with proper literature review and comments, and not placed as breakthrough or ideas. They shall leave the suggestions for the conclusions, which could be changed to final considerations and then, list some of the author's ideas of what it needs to be discovered. Doing this, the authors separate what is proved science and what is pure speculation.
The reviewer is right, that this section may be disorganized, as some topics just point out an idea (mainly if previously has been described/discussed) but one of the most important function of a review is to organize, evaluate, synthesize the literature, but to to identify research gaps and recommend new research areas in the future, this is what section 4 intended. Following the reviewers advise we have converted it into future considerations, and we have given a more “paragraph” format to the section. Some ideas did not have citations as they had been previously discussed, but as the reviewer mentioned, the old structure was somehow unbalanced, we think that with this new structure now is more balanced.
Round 2
Reviewer 3 Report
The authors successfully answered all my comments and revised the manuscript accordingly. I believe that know it can be published. I have two very minor comments and the I don´t need to see the revised version:
L114: first biggest agricultural sector in the EU
First biggest seams odd to me, please rephrase.
L378: libitum (Pisoni et al., 2020).
Incorrect citation style
Author Response
L114: first biggest agricultural sector in the EU
First biggest seams odd to me, please rephrase.
In current paper it corresponds to Line 159- it has been changed to "largest"
L378: libitum (Pisoni et al., 2020).
Incorrect citation style
In the current paper it corresponds to Line 975-it has been changed to [54]